# MaskDGNets: Masked-attention guided dynamic graph aggregation network for event extraction

Guangwei Zhang[1], Fei Xie[2]*, Lei Yu[2]

1 Didi Infinite Technology Development Co., LTD., Beijing, China, 2 Shanghai University of Finance and Economics, Shanghai, China

* 1281183048@qq.com

**Data Availability Statement:** This manuscript's minimal data set is available via FigShare at the following URLs: DuEE: https://figshare.com/ articles/dataset/DuEE1_0_zip/27283248?file= 49937835 CCKS2020: https://figshare.com/

## Abstract

Considering that the traditional deep learning event extraction method ignores the correlation between word features and sequence information, it cannot fully explore the hidden associations between events and events and between events and primary attributes. To solve these problems, we developed a new framework for event extraction called the masked attention-guided dynamic graph aggregation network. On the one hand, to obtain effective word representation and sequence representation, an interaction and complementary relationship are established between word vectors and character vectors. At the same time, a squeeze layer is introduced in the bidirectional independent recurrent unit to model the sentence sequence from both positive and negative directions while retaining the local spatial details to the maximum extent and establishing practical long-term dependencies and rich global context representations. On the other hand, the designed masked attention mechanism can effectively balance the word vector features and sequence semantics and refine these features. The designed dynamic graph aggregation module establishes effective connections between events and events, and between events and essential attributes, strengthens the interactivity and association between them, and realizes feature transfer and aggregation on graph nodes in the neighborhood through dynamic strategies to improve the performance of event extraction. We designed a reconstructed weighted loss function to supervise and adjust each module individually to ensure the optimal feature representation. Finally, the proposed MaskDGNets framework is evaluated on two baseline datasets, DuEE and CCKS2020. It demonstrates its robustness and event extraction performance, with $F_1$ of 81.443% and 87.382%, respectively.

## 1 Introduction

With the rapid development of the Internet and media, event extraction has been widely considered an essential information extraction task [1]. Its primary purpose is obtaining all the information from complex, unstructured, or structured texts, sentences, and documents. All events included. Effective event extraction can significantly support the implementation of

**Funding:** The author(s) received no specific
funding for this work.

**Competing interests:** The authors have declared
that no competing interests exist.

many downstream tasks, such as knowledge graph construction [2], social event detection [3], event prediction [4], and intelligence and business economic formulation [5]. Then, as information grows exponentially, it will be difficult to fully capture the hidden correlations between events using simple machine learning or deep learning methods. Therefore, how to better capture the deep local sum of events in texts or sentences Global semantics and relevant contextual details to improve these correlations in events have become a research hotspot.

In recent years, some researchers have designed various solutions to improve the accuracy of event extraction by obtaining the detailed semantics of events in texts or sentences and the details of many attributes, such as entities and time associated with events. Early event extraction methods primarily relied on expert experience to create manual features as primary inputs. They defined them as classification behaviors while using simple machine learning methods such as random forests, support vector machines, logistic regression, and decision trees as classifiers to improve the performance of event extraction [6–8]. Although these simple methods effectively improve the efficiency and accuracy of event extraction, they still require manual participation in feature design and screening. They are heavily dependent on researchers and expert experience. There is much room for improvement in accuracy, and it isn't easy to adapt to large-scale data. To address these limitations, many researchers have applied traditional deep learning technology to this task, relying on the powerful self-learning ability of neurons in the network and the ability to obtain deep discriminant features while reducing manual participation in feature screening and design and improving the performance of event extraction. For example, using a deep convolutional neural network [9] to extract local spatial details of events in texts or sentences and using a recursive neural network to model global and contextual information, but due to the limitation of the receptive domain, it is difficult to capture the details of event attributes fully. Neurons in recurrent neural networks (RNNs) [10] must improve their interpretability. At the same time, the network is prone to problems such as fitting and gradient explosion and cannot effectively establish long-term dependencies. There is still a lot of room for improvement in event extraction performance. The Long-short term memory networks (LSTMs) [11] effectively alleviate the problems of RNNs and can describe text events from both positive and negative directions. Subsequently, some researchers considered the relationship between events and essential attributes such as entities, roles, and time. Graph methods, transformers, and joint extraction methods were widely used [12, 13]. Graph methods mainly establish connections between events and these essential attributes and achieve accurate descriptions of events through the convergence and transmission functions between nodes. Although these extraction methods using graph networks have achieved encouraging results, only the importance of local features is considered in constructing topological maps. In contrast, the importance of global information, contextual semantics, and the ambiguity and correlation between the same entity or other attribute information in different events are ignored.

To address these issues, we propose the MaskDGNets event extraction framework, which aims to model the semantic details of events and related essential attributes. First, sentences or text sequences are mapped, effective local, global, and contextual semantics are obtained in a unified low-dimensional space, and complementary and dependent relationships are formed between them. Secondly, this feature information is processed and refined, the representation of salient features is highlighted, and redundant information is reduced. In addition, dependencies are established between events and essential attributes, thereby improving the performance of event extraction by mining hidden correlations between them.

This study main contributions are as follows.

1. A masked attention-guided dynamic graph aggregation network (MaskDGNets) for event extraction is proposed, aiming to obtain effective semantic representation to improve the

performance of event extraction. A spatially enhanced bidirectional independent recurrent neural network enriches global and contextual semantics while ensuring local spatial details. In addition, individual character features are embedded into word vectors to avoid the ambiguity of the exact character representation in different words. At the same time, without adding additional auxiliary information, when the word vector feature representation is insufficient, the character feature will be represented as the main feature, which is beneficial to the description of event types in sentence sequences.

2. A masked attention-guided dynamic graph aggregation module is designed to mine the hidden associations between events and events and between events and primary attributes. At the same time, the event node and attribute node features in the neighborhood are aggregated and transferred through a dynamic aggregation strategy. In addition, the masked attention module can refine the word vector features and sequence information and establish interactions between them to strengthen the representation of features.

3. Developed a reconstructed weighted loss function to supervise and adjust each module separately, prompting the MaskDGNets event extraction framework to learn the optimal features to improve event extraction performance. Finally, we conducted evaluation and verification on two sets of baseline datasets, both of which achieved exemplary performance and robustness.

The rest of this study is organized as follows: Section 2 introduces a series of related works on event extraction. Section 3 elaborates on the working principle of the proposed MaskDG-Nets event extraction framework and the functions of each module. Section 4 provides many experimental results and analyses, as well as the ablation details of each module. Section 5 gives the conclusion and the following research plan.

## 2 Related works

Early event extraction tasks primarily focused on using simple machine-learning methods to operate on handcrafted features. For example, considering that an event may contain multiple entities or other relationships, Saha S et al. [14] proposed a supervised classification method of support vector machine. This method uses statistical and linguistic features to represent the various morphological, syntactic, and contextual information of candidate biomolecule trigger words and classifies these identified events into nine predefined categories. Saigal P et al. [15] introduced the least squares strategy in the support vector machine to adapt to the multi-class event relations in data to achieve better classification results. At the same time, the feature set was constructed as a word frequency-inverse document frequency matrix to obtain a representative vector for each document. Shanmugavadivel K et al. [16] mapped the text into digital features as training samples for the machine learning model to automatically identify and extract noteworthy events reported in a collection of articles. They used the random forest method to classify these events effectively. To understand the temporality of key sentences in texts and distinguish between contextual information and valuable details, García-Méndez S et al. [17] combined natural language processing and machine learning techniques to detect the temporality of financial at the discourse level and used complex features such as syntactic and semantic dependencies to extract related events. To detect fake events and alleviate the problem of data imbalance, Hakak S et al. [18] proposed an integrated classification model to extract important detail features from fake datasets. The integrated model was composed of three popular machine learning models, such as decision trees, random forests, and extra tree classifiers, and it classified the extracted features. Although these methods can improve event extraction performance, they require manual participation in feature screening and threshold

setting, are heavily dependent on expert experience, and are difficult to apply on a large scale. At the same time, the simple machine learning method has low extraction accuracy and is time-consuming, labor-intensive, and expensive.

In recent years, with the continuous development of deep learning technology, researchers have designed many methods to improve the errors caused by manual features. For example, to better encode contextual semantics and external background knowledge, Li D et al. [19] developed a knowledge base-driven tree-structured extended short-term memory network (Tree-LSTM) framework, which mainly relies on structure to capture a wide range of contextual details and obtain entity attributes from external ontology through entity links, thereby improving the performance of event extraction. Considering the differences and interactions between events, Li Q et al. [20] designed an event-driven LSTM model to alleviate these limitations and reduce the loss of detailed information. Songklang K et al. [21] used a bidirectional long short-term memory (Bi-LSTM) model to study event detection and analysis in Thai. They achieved an accuracy of 73.41% and 81.71% in event type classification and event component identification, respectively. Considering that the existing event extraction methods are primarily focused on rules or pattern strategies and have poor generalization ability, Guo L et al. [22] proposed a deep neural network method, that is, using bidirectional LSTM to pre-train word embedding in feature fields and using conditional random fields to capture the interaction between triggers and parameters. Zeng Y et al. [23] considering that previous event extraction methods were highly dependent on complex feature engineering and complex natural language processing (NLP) tools, proposed the language specificity problem in Chinese event extraction and then combined the convolutional bidirectional LSTM neural network of LSTM and CNN to capture sentence-level and vocabulary-level information simultaneously without any manual features. To better obtain the contextual semantics in the event text and establish practical long-term dependencies, Banerjee S et al. [24] proposed a transformer headline event extraction method, which uses the sentence extractor of the deep neural network and the summary extractor of the transformer to work at one time to generate event titles, where the sentence extraction task is positioned as a binary classification task. The transformer forms the event title from the extracted sentence. These methods mostly use traditional deep learning models, which could better capture deep semantics and detail modeling. At the same time, they rely on additional auxiliary information to improve the extraction performance. Therefore, graph and joint extraction methods have attracted the attention of many researchers and scholars and have been applied to event extraction tasks.

Influenced by the successful application of graph neural networks in many fields, Lee M et al. [25] proposed a method to effectively use graph convolutional networks and pruned dependency parse trees to extract events from commodity more effectively. They used the embedding features of a BERT-masked language model as the model's training information. Wu X et al. [26] developed a character-level Chinese event extraction framework based on a graph attention network (CAEE) due to the poor performance of Chinese text caused by the mismatch between word segmentation and labeling. A new framework was built based on the sequence labeling model, which enhanced word information by further incorporating word dictionaries into character representation to exploit the interdependence between event triggers and parameters. They also constructed a word-character-based graph network using grammatical shortcut arcs of dependency parsing. Wan Q et al. [27] designed a new framework for document open event extraction to obtain enhanced dependency structures with strong encoding capabilities. At the same time, it considers the order of sequential structures and associates ancestor nodes and descendant nodes to establish a bidirectional dependency parsing graph. It uses node information to strengthen the aggregation of node features in the graph attention model. Then, feature information such as semantics, syntax, and POS

cooperate to improve the performance of event extraction. To obtain practical event details, Orr J W et al. [28] embedded the syntactic dependency structure into the GRU model of the directed graph, which improved the event classification performance. Due to the possibility of overlapping entities in the event extraction process, Hei Y et al. [29] proposed a cascade-level framework for event color correction label graph enhancement to address these limitations. The framework reduces errors between subtasks by gradually passing smooth features to downstream subtasks. At the same time, the event-to-event and role-to-role association information is merged into the label space of event type and parameter role, which helps to assign overlapping elements to multiple labels with approximate feature spaces. Although these graph neural network methods strengthen the interaction of entities and attribute information in events through node aggregation and transmission capabilities, they often ignore the errors caused by the same entities in different events. At the same time, the hidden relationships between events still need to be fully considered in the event modeling process.

Li Q et al. [30] achieve joint extraction of trigger words and arguments by combining global text features and deep learning. Wang H et al. [31] proposed a joint constraint learning framework to model event-event relations, enforcing logical constraints within and across multiple temporal and sub-event relations by converting these constraints into distinguishable learning objectives. Zhang Z et al. [32] consider that abstract meaning representation (AMR) has similar goals to information extraction, converting natural language text into structured semantic representations. Therefore, to exploit this similarity, a new AMR-guided framework for joint information extraction is proposed to discover entities, relations, and events with the help of a pre-trained AMR parser, a semantic graph aggregator with abstract meaning representation is adopted to enable candidate entities and event trigger nodes to collect neighborhood information from the AMR graph to pass messages between related knowledge elements. At the same time, the graph decoder is guided in the order of the AMR hierarchy to extract the prior knowledge of elements in the event. Li H et al. [33] developed a prompt-based graph model for free event extraction (PGLEE). This model obtains candidate triggers and parameters and constructs a heterogeneous event graph to encode the structure within and between events, thereby improving the performance of event extraction. Although the joint learning method has certain advantages in theory, its model design is complex, and it needs to consider more parameters and multi-threaded output, which has particular difficulties in actual operation.

## 3 Proposed framework of MaskDGNets

This section first gives a basic overview of the proposed MaskDGNets event extraction framework and briefly introduces each module of internal structure, working principle, and training steps.

### 3.1 Overview

The overall structure of the proposed MaskDGNets for event extraction framework is shown in Fig 1. The framework mainly consists of four parts: a word vector embedding module (WVEM), a sequence semantic extraction module (Bi-Indrnn+), a masked attention embedding module (MaskAM), and a dynamic graph aggregation module (DGAM). The word vector embedding module (WVEM) aims to obtain useful word vectors and character features and form interactive and complementary relationships. The sequence semantics extraction module (Bi-Indrnn+) mainly models the entire input sequence and extracts global and contextual semantics that benefit event representation while retaining local spatial features to the maximum extent. The mask attention embedding module (MaskAM) mainly improves the representation of events by fusing sequence features and word vector features. It uses a weight

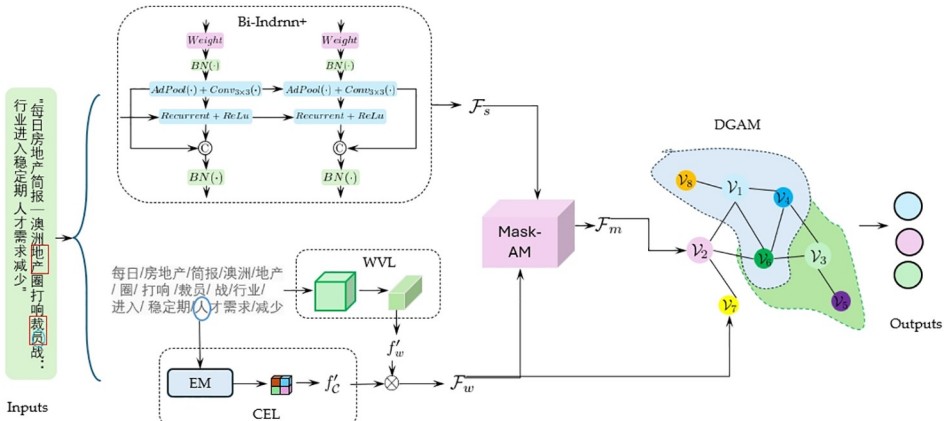

**Fig 1. The overall network structure of MaskDGNets.** Where, 'Bi-Indrnn+' indicates an improved bidirectional independent recurrent neural network module. 'WVL' indicates the word vector layers. 'CEL' indicates the character embedding layer. 'EM' indicates the embedding module. 'DGAM' indicates the dynamic graph aggregation module. 'MaskAM' indicates the mask attention module. $\{\mathcal{F}_w, \mathcal{F}_s, \mathcal{F}_m\}$ indicates word vector feature with word and characters, the sentence sequence features and fusion feature by mask attention module. $\{f'_w, f'_\mathcal{C}\}$ indicates the word feature and character feature. $\otimes$ indicates the feature matrix multiplication. $BN(\cdot)$ indicates the batch normalization operation. *Weight* indicates the weight matrix. *Recurrent* represents the recurrent unit. *ReLu* represents the activation function. $AdPool(\cdot)$ represents the adaptive pooling operation. © represents feature concatenation. $Conv_{3\times3}(\cdot)$ indicates $3 \times 3$ convolution operation. $\{\mathcal{V}_1, \ldots, \mathcal{V}_8\}$ indicates the graph nodes, where $\{\mathcal{V}_1, \mathcal{V}_2, \mathcal{V}_3\}$ indicates the event node in the sentence, $\{\mathcal{V}_4, \ldots, \mathcal{V}_8\}$ represents the basic attribute node of the event in the sequence *S*.

allocation strategy to highlight salient features and refine them at the same time. The dynamic graph aggregation module (DGAM) establishes dependencies between events and events and between events and primary attributes. It mines hidden correlations between them through dynamic node aggregation and transfer functions to improve the performance of event extraction.

## 3.2 WVEM

This module mainly consists of a word vector mapping layer (WVL) and a single-character embedding layer (CEL). The word vector mapping layer (WVL) mainly maps words in text or sentences to low-dimensional space to better represent word vector features. Single characters are essential components of words, and different characters have different semantics or contexts in words or phrases. To better learn word vector features and sentence representation, we map characters separately to reduce the semantics of the same character in different phrases, which helps to improve the representation of keywords in sentences.

For the word vector mapping layer (WVL), assume that the given Chinese event sequence is $S = \{\mu_1, \mu_2, \cdots, \mu_N\}$, where $\mu_i \in S$ represents the $i^{th}$ word or phrase in the sentence sequence. We use the commonly used word2vec pre-trained model to map these words or phrases to obtain word vector features $f_w$, where the number of word2vec [34] iterations is set to 10 and the sliding window is set to 7. Then, we squeeze these output word vector features $f_w$ to alleviate redundant information and obtain effective word features $f'_w$. The specific operation is shown in the Eqs (1) and (2).

$$f_w = word2ve(\mu_1, \cdots, \mu_i, \cdots \mu_N) \tag{1}$$

$$f'_w = AvgPool(Conv_{\times1}(f_w)) \tag{2}$$

Where $\mu_i$ indicates $i^{th}$ word. $N$ indicates the length of the sentence, namely, the number of words in the sentence sequence $S$. $Conv_{1\times1}(\cdot)$ indicates $1 \times 1$ convolution operation. $AvgPool(\cdot)$ indicates the average pooling operation. This squeezing method reduces redundant information while retaining the details of the word vector to the maximum extent. And improves the representation of the word vector features.

For the character embedding [35] layer (CEL), the Chinese sequence is $S$. Each $S$ consists of $N$ words, and the word $\mu$ consists of $M$ single characters, defined as $\mu_i = \{C_1, \ldots, C_j, \ldots, C_M\}$. We use a simple word embedding layer to embed it to obtain the character feature of $f_C$. Then, to reduce the reuse of redundant information and highlight the differences in the representation of the same character in different words, we use the maximum pooling layer to operate it and obtain the practical character feature $f_C'$. The specific operation is shown in the Eqs (3) and (4).

$$f_C = EM(C_1, \cdots, C_j, \cdots, C_M) \tag{3}$$

$$f_C' = MaxPool(f_C)) \tag{4}$$

Where $EM(\cdot)$ indicates embedding module. $MaxPool(\cdot)$ indicates max pooling operation. To make the word vector feature better represent the event extraction task, we interactively learn between the character feature $f_C'$ and the initial word vector feature $f_w'$. This method can make the other feature play a key role when one feature is insufficient, and no additional information is needed. The specific operation is shown in the Eq (5).

$$\mathcal{F}_w = f_w' \otimes f_C' \tag{5}$$

Where $\otimes$ indicates matrix multiplication.

## 3.3 Bi-Indrnn+

Recurrent neural networks (RNNs) have been proven effective in obtaining helpful sequence information for text or sentence sequence tasks. However, RNNs are prone to problems such as fitting and gradient explosion, and it is challenging to establish practical long-term dependencies. Compared with traditional RNNs, independent recurrent neural networks (Indrnn) [36] can effectively solve the gradient vanishing and gradient explosion problems and thoroughly learn long-term dependencies because neurons are independent of each other and can transmit information across layers. At the same time, with the help of non-saturated activation functions such as $ReLu$, IndRNN becomes very robust after training, and by stacking multiple layers of IndRNN, a deeper network than the existing RNN can be constructed. Therefore, we designed an improved bidirectional independent recurrent neural network (Bi-Indrnn+), combining a set of $3 \times 3$ convolution and adaptive pooling operations with independent recurrent units. We squeezed the input feature information to reduce the number of calculation parameters and obtain the sequence details of sentences or texts.

For a sentence sequence $S = \{\mu_1, \mu_2, \cdots, \mu_N\}$, the forward indrnn + learns each word $\mu_i$ in the sequence $S$ and encodes the sequence $S$ in a forward direction and marked as $\overrightarrow{H}$. The back-to-front learning can encode the sequence $S$ in a reverse direction and marked as $\overleftarrow{H}$. This method of encoding the sequence $S$ from different directions and angles is beneficial to the representation of context and global information, specifically, as shown in the Eqs (6) and (7).

$$\overrightarrow{H} = \sigma(\overrightarrow{\mathcal{W}}\overrightarrow{\chi_S} + \overrightarrow{\partial} \odot \overrightarrow{H}_{t-1} + \overrightarrow{b}) \otimes \overrightarrow{\chi_S} \tag{6}$$

$$\overleftarrow{H} = \sigma(\overleftarrow{\mathcal{W}}\overleftarrow{\chi_S} + \overleftarrow{\partial} \odot \overleftarrow{H}_{t-1} + \overleftarrow{b}) \otimes \overleftarrow{\chi_S} \tag{7}$$

Where $\overrightarrow{\mathcal{W}}, \overrightarrow{\partial}, \overleftarrow{\mathcal{W}}$ and $\overleftarrow{\partial}$ represent the weight matrices in the forward and reverse directions, respectively. $\overrightarrow{H}_{t-1}$ and $\overleftarrow{H}_{t-1}$ represent the output features at time $t-1$ in the forward direction. $\otimes$ represents matrix multiplication. $\odot$ represents Hadamard products. $\overrightarrow{\chi_S}$ and $\overleftarrow{\chi_S}$ represent the sequence features of the forward and reverse inputs, and their calculations are shown in the Eq (8).

$$\overrightarrow{\chi_S} = AdPool(Conv_{3\times3}(X_S)) \tag{8}$$

Where $Conv_{3\times3}(\cdot)$ indicates $3 \times 3$ convolution operation. $AdPool(\cdot)$ indicates the adaptive pooling operation. $X_S$ indicates the input sequence. Finally, the sequence information of the square in different directions is merged to obtain the global and context sequence information $\mathcal{F}_s$ with different directions and angles. The calculation of $\mathcal{F}_s$ is shown in the Eq (9).

$$\mathcal{F}_s = BN(\overrightarrow{H} © \overleftarrow{H}) \tag{9}$$

Where, © indicates the feature concatenation operation. $BN(\cdot)$ indicates the batch normalization layer.

## 3.4 MaskAM

Adequate contextual semantics help mprove the representation of global features. It can increase the association between keywords and event types in sentences or documents, reduce the errors caused by the same entities or keywords in different event types, and better refine local details. The specific steps are as follows. The local features of the input sentence or text can effectively update the query features and collect context information by paying attention to each query [37]. At the same time, these local features are refined. The specific calculation mask feature of $\mathcal{F}_m$ is shown in the Eq (10).

$$\begin{aligned} \mathcal{F}_m \quad &= \sigma'(\mathcal{M} + QK^T)V + [\mathcal{F}_w \oplus \mathcal{F}_s] \\ &= \sigma'(\mathcal{M} + f_Q(\mathcal{F}_w \oplus \mathcal{F}_s)K^T)V + [\mathcal{F}_w \oplus \mathcal{F}_s] \end{aligned} \tag{10}$$

Where, $\mathcal{F}_w$ indicates the word feature. $\mathcal{F}_s$ indicates the sentence features. $f_Q(\cdot)$ indicates the linear transformations. $K$ and $V$ indicates by linear transformations feature of $f_K(\cdot)$ and $f_V(\cdot)$. In addition, the attention mask $\mathcal{M}$ at feature local $i, j$ is define in the Eq (11).

$$\mathcal{M} = \begin{cases} 0, \mathcal{M}(i,j) = 1 \\ -\infty, others \end{cases} \tag{11}$$

In summary, this interactive method can fully capture the interactive information and position correlation between events and other attributes in sentences or texts while reducing the transmission of redundant information flow.

## 3.5 DGAM

A graph convolutional neural network [38–40] is a network that can perform convolution operations directly on topological graphs. At the same time, it can learn node representation by modeling the complex structure and rich semantics of the graph through nodes' aggregation and transmission capabilities. Therefore, to improve the description of event types or behaviors in sentences or texts and to better mine the hidden associations between events and events and between events and other attributes, we constructed a sizeable topological graph whose graph nodes are mainly two types of nodes, event types and essential attributes of

events, to capture the associations between them, and further improve the representation of local features, global features, and contextual semantics, thereby improving the overall performance of the network.

The topological graph we constructed is defined as $\mathcal{G} = (\mathcal{V}, \mathcal{E})$, where $\mathcal{V}$ and $\mathcal{E}$ represent the node set and edge set, respectively. All event types contain $\mathcal{N}$ event nodes and $\mathcal{U}$ event attribute nodes and satisfied with $|\mathcal{V}| = \mathcal{N} + \mathcal{U}$. It is worth noting that the nodes in the topological graph are self-loop nodes, and the nodes themselves have valid feature information. The initial graph convolution definition of $\mathcal{F}_\mathcal{G}$ is shown in the Eq (12).

$$\mathcal{F}_\mathcal{G} = \sigma\left(\tilde{D}^{-\frac{1}{2}} \cdot \tilde{A} \cdot \tilde{D}^{-\frac{1}{2}} \cdot \mathcal{X} \cdot W_\mathcal{G}\right) \tag{12}$$

Where $A$ represents the adjacency matrix. $D$ represents the degree matrix. $W_\mathcal{G}$ represents the weight matrix of the graph. $\mathcal{X}$ represents the input features.

Considering that the performance of graph convolutional networks is heavily dependent on the graph structure, there is redundant information when constructing the topological graph, and there may be missing edges between events and events and between events and essential attribute nodes, resulting in decreased event extraction performance. Therefore, we designed a dynamic graph aggregation network to obtain the optimal graph structure that adaptively adds or deletes redundant edges. To better aggregate the feature information of nodes in the neighborhood, we use vector similarity to measure whether there are associated edges between different nodes. The adjacency matrix $A$ is shown in the Eq (13).

$$A_{i,j} = \begin{cases} 0, \, others \\ 1, \mathcal{V}_i = \mathcal{V}_j \\ log_2 \dfrac{\varphi(\mu_i, \mu_j)}{\varphi(\mu_i) * \varphi(\mu_j)}, \{\mu_i \neq \mu_j\} \in \{\mu_1, \cdot, \mu_N\} \\ \dfrac{S_i \cdot S_j}{||S_i|| ||S_j||}, S_i \neq S_j \end{cases} \tag{13}$$

Where, $S_i$, $S_j$ indicates the $i^{th}$ and $j^{th}$ sentence sequence. $\mu_i$, $\mu_j$ represents the $i^{th}$ and $j^{th}$ word. The initial graph matrix $H_\zeta$ is shown in the Eq (14).

$$H_\zeta = \mathcal{F}_\mathcal{G}(\{\mathcal{F}_m, \mathcal{F}_w\}; A) \tag{14}$$

Where $\mathcal{F}_m$ indicates the mask feature by MaskAM. $\mathcal{F}_w$ indicates the feature of word in sentence. That is when a node in the topology graph is an attribute node, its node feature is $\mathcal{F}_w$. We modified adaptive adjacency matrix $\tilde{A}$ by initial graph mattrix $H_\zeta$ at next graph layers. And the $\tilde{A}$ and the $l^{th}$ output graph feature $H_\zeta^{(l)}$ is shown in the Eqs (15) and (16).

$$\tilde{A} = \sigma^*(H_\zeta \cdot \tilde{W} \cdot H_\zeta^\top) \tag{15}$$

$$H_\zeta^{(l)} = \sigma\left(\tilde{D}^{-\frac{1}{2}} \cdot \tilde{A} \cdot \tilde{D}^{-\frac{1}{2}} \cdot H_\zeta^{(l-1)} \cdot W_\mathcal{G}\right), l \geq 2 \tag{16}$$

Where $\tilde{W}$ indicates a learnable matrix. $l = 1$ indicates $H_\zeta^0 = \mathcal{F}_m$

In summary, the designed dynamic graph aggregation network captures the graph structure, enhancing the dependencies between events and events, events, and other attributes. It strengthens the interactions between them, further improving the event extraction performance.

### 3.6 Reconstructed loss function

We designed a weighted reconstruction loss function to enable the network to obtain the optimal feature representation and achieve overall framework regulation. We used different loss functions for each stage to perform separate supervision to prevent the network from falling into the local optimum. The weighted reconstruction loss function is shown in the Eq (17).

$$\begin{cases} L_{total} = L_m + \lambda \cdot L_a + \alpha \cdot L_b + \beta \cdot (L_c \otimes \gamma) \\ \alpha = 0.15, \lambda = 0.2, \beta = 0.4 \end{cases} \tag{17}$$

Where, $\alpha$, $\lambda$ and $\beta$ represent learnable balancing factors. $\gamma$ represents the class weight matrix. $\otimes$ indicates the matrix multiplication.

For the primary loss function of $L_m$, we first input the feature information aggregated by the dynamic graph aggregation module into the fully connected layer and use the softmax function to predict the distribution probability. The specific operation is shown in the Eqs (18) and (19).

$$L_m = -\sum_{s=1}^{N_s} \sum_{\mu=1}^{N_\mu} \omega_i log \sigma'(W \cdot FC(O_\zeta) + b) \tag{18}$$

$$\omega_i = \frac{(Mid(N_w^1, \cdots, N_w^m) + Max(N_w^1, \cdots, N_w^m))}{2 \cdot N_c^i} \tag{19}$$

Where, $FC(\cdot)$ indicates the fully connected operation. $\sigma'(\cdot)$ indicates the function of SoftMax. $b$ indicates the bias weights. $N_s$ indicates te number of sentences. $\omega_i$ indicates the weight of $i^{th}$ label. $N_c^i$ indicates the $i^{th}$ class label.

For the 'MaskAM' and 'Bi-Indrnn+' modules, we input the extracted feature information into a adaptive pooling layer for feature compression and then input it into the fully connected layer to achieve separate supervision of these modules. $L_a$ and $L_b$ represent the loss functions of MaskAM and Bi-Indrnn + is the Focal loss of multi-classification, as shown in the following Eqs (20) and (21).

$$L_a = -\sum \alpha_t \cdot \Theta \cdot (1 - Y_t)^\eta log(Y_t) \tag{20}$$

$$Y_t \in \{\sigma'(FC(AdPool(O_{Bi+}))), \sigma'(FC(AdPool(O_{Ma+})))\} \tag{21}$$

Where, $FC(\cdot)$ indicates the fully connected operation. $AdPool(\cdot)$ indicates the adaptive pooling operation. $\alpha_t$ is a list containing the weights of each category. $\Theta$ is the one-hot vector of the current sample.

For the dynamic graph aggregation module (DGAM), we input the graph nodes' embedding information into the SofMax classifier, whose loss function defines the cross-error of event types. AA is as shown in the Eq (22).

$$L_c = -\sum_{d \in S_d} \sum_{f=1}^{F} S_d \cdot ln(\sigma'(H_\zeta^{(l)})) \tag{22}$$

Where, $H_\zeta^{(l)}$ indicates the output feature of dynamic graph aggregation module with $i^{th}$ layers. $S_d$ indicates the event sentence index with label. $F$ indicates the feature dimension. $\sigma'(\cdot)$ indicates the function of SoftMax.

In summary, the weighted reconstruction loss function we proposed balances different types of events and different attributes of the same type of events. It adopts this separate

supervision strategy to enhance the representation performance of hidden features and discriminative features in different modules, thereby improving the overall performance of the proposed MaskDGNets event extraction framework.

**Algorithm 1:** The process of event extraction by our develop MaskDGNets

**Input:** Given a sentence event sequence $S = \{\mu_1, \cdots, \mu_N\}$, $\mu_i = \{\mathcal{C}1,...,\mathcal{C}_M\} \in S$ represents the word in the sentence, and $\mathcal{C}_j$ represents the $j^{th}$ character in each word. $\mathcal{F}_w, \mathcal{F}_s$, and $H_\zeta$ are word features, sentence sequence features, and fused graph features, respectively. The reconstructed weighted loss function of $L_{total}$ and the optimizer *AdamW* supervise the event extraction framework.;

**for** $e$=0 to $e$=Max **do**

$[f'_w, f'_\mathcal{C}] \xleftarrow[MaxPool(\cdot)]{\frac{Conv_{1\times1}(\cdot) + AvgPool(\cdot)}{}} [f_w, f_\mathcal{C}] \xleftarrow[Eq.1]{Eq.3} [\mu_i, \mathcal{C}_j];$

$[\mathcal{F}_w, \mathcal{F}_s] \xleftarrow[Eq.5]{Bi-Indrnn+} [f'_w, f'_\mathcal{C}, S];$

$\mathcal{F}_m \xleftarrow[MaskAM]{Eq.10, Eq.11} [\mathcal{F}_w, \mathcal{F}_s];$

$H_\zeta \xleftarrow[DGAM]{Eq.14-Eq.16} [\mathcal{F}_m, \mathcal{F}_w, A] \xleftarrow{Eq.13} [\mathcal{V}_{i,j} \subset \{\mu_i, \mathcal{C}_j, S\}];$

This final optimization is via Eq 17 to Eq 22;

**end**

**output:** optimization the training during by *AdamW* and total loss function of $L_{total}$.

## 4 Experimental and discussion

This section introduces the experimental data source, environment, detailed parameters of the proposed MaskDGNets framework, and the evaluation indicators. Secondly, we compare with the current advanced event extraction methods and provide corresponding analysis and experimental results. Finally, a series of ablation experiments are conducted under the same experimental conditions to demonstrate whether each component in the proposed framework plays a positive role in the model's overall performance. In addition, various discussions are given to demonstrate the proposed MaskDGNets framework's overall performance intuitively.

### 4.1 Datasets preparation

**DuEE.** [41] Baidu released this dataset, which is mainly used for Chinese event extraction. It contains 65 event types, including marriage, resignation, earthquake, winning, and competition, and 17,000 sentences with event information. In addition, these sentences come from Baidu information flow information text. Compared with traditional information, the text has a higher degree of freedom of expression, and the difficulty of event extraction is also more incredible.

**CCKS2020.** This dataset comes from the CCKS2020 "Chapter-level event subject and element extraction for the financial field" academic evaluation task, jointly provided by Ant Group and the Institute of Automation of the Chinese Academy of Sciences. It contains 72,515 annotated data, including various event types such as significant compensation and equity pledges. It is worth noting that a text may have multiple event types, or the same event type may have multiple event subjects. In addition, in order to ensure the consistency and correctness of the experiment, we divide these data into three parts: training, verification, and testing, with a ratio of 3: 1: 1. At the same time, to alleviate the errors caused by data quality, we will use a unified word segmentation tool and stop words, as well as data cleaning methods.

### 4.2 Related settings

**4.2.1 Evaluation metrics.** . To ensure the fairness and smooth progress of all experiments, we measure all event extraction methods using three commonly used evaluation indicators:

precision (P), recall (R), and F-score ($F_1$). The specific calculation is shown in the equation.

$$P = \frac{TP}{TP + FP} \tag{23}$$

$$R = \frac{TP}{TP + FN} \tag{24}$$

$$F_1 = \frac{2 \cdot TP}{2 \cdot TP + FN + FP} \tag{25}$$

Where, $TP$, $TN$, $FP$ and $FN$ indicate true positive, true negative, false positive, false negative, respectively.

**4.2.2 Parameters settings.** . The proposed MaskDGNets event extraction framework uses Adamw as the optimizer in the training phase, with a decay rate of $1e - 5$, a learning rate of $1e - 4$, and 200 training times. To prevent the network from falling into a local optimal state or fitting, the cosine annealing algorithm is used to adjust the learning rate dynamically, and an early stopping strategy is adopted; namely, if the network has problems such as fitting, the iteration number is 20. In addition, all experiments in this study were conducted in a unified environment; that is, python3.8.13 was used as the programming language, and training and testing were performed on a 4-card $RTXA6000$ GPU machine. Other required environments include deep learning libraries such as cu117, torch1.13.0+cu117, Numpy, and Pandas.

## 4.3 Comparison with other state-of-the-art methods

To demonstrate the proposed MaskDGNets event extraction framework with effectiveness and superiority, we compare it with the current advanced event extraction methods on various open-source datasets and give the corresponding analysis and experimental results. The experimental results of different methods are shown in Table 1.

From Table 1, we can get the following conclusions.

(1) On the two baseline data sets of *DuEE* and *CCKS*2020, our proposed MaskDGNets event extraction framework achieved the optimal performance, and $F_1$ was 81.443% and 87.382%, respectively. The possible reason is that, on the one hand, the spatial and sequence details in the sentence are refined through the mask attention module, and the representation

**Table 1. Experimental results of different event extraction methods on the DuEE and CCKS2020 datasets.**

| Models | DuEE | | | CCKS2020 | | |
|---|---|---|---|---|---|---|
| | $P$ | $R$ | $F_1$ | $P$ | $R$ | $F_1$ |
| JMEE | 55.106 | 52.599 | 53.823 | 60.323 | 69.095 | 64.411 |
| DMCNN | 65.104 | 53.539 | 58.758 | 61.133 | 71.948 | 66.101 |
| Bi-LSTM | 67.129 | 58.437 | 62.482 | 62.86 | 73.092 | 67.591 |
| JRNN | 70.724 | 64.145 | 67.274 | 63.478 | 73.459 | 68.105 |
| BERT-CRF | 72.212 | 65.412 | 68.644 | 68.118 | 75.279 | 71.519 |
| PLMEE | 74.266 | 70.748 | 72.464 | 69.113 | 81.537 | 74.813 |
| Multi-GAT | 76.838 | 72.375 | 74.54 | 70.813 | 81.54 | 75.799 |
| GCN-ED | 78.175 | 72.693 | 75.334 | 71.089 | 82.128 | 76.211 |
| EE-GCN | 79.632 | 77.649 | 78.628 | 71.296 | 83.829 | 77.056 |
| **MaskDGNets** | **84.363** | **78.718** | **81.443** | **86.616** | **88.162** | **87.382** |

Bold font indicates the optimal performance.

of salient features is highlighted by assigning different weights; at the same time, the improved two-way independence is adopted Recurrent neural network models contextual semantics from both positive and negative directions and strengthens long-term dependencies. On the other hand, the dynamic graph aggregation module alleviates the differences between hierarchical graph structures and attribute nodes. It enhances the interaction between events and related attribute information through encoding and dynamic aggregation, improving the overall framework performance.

In addition, single-character features are embedded into word vectors to avoid ambiguity in the exact character representation in different words. When word vector features are insufficient without adding auxiliary information, character features are used as the main features for representation, which is beneficial for describing event types in sentence sequences. At the same time, a new weighted loss function is used to perform separate supervised learning and optimization adjustments on each module so that the proposed MaskDGNets event extraction framework can learn the optimal features, thereby improving event extraction performance.

(2) Graph methods are more competitive on both datasets than traditional deep learning methods. On the DuEE dataset, EE-GCN of $F_1$ is 6.164% and 11.454% higher than *PLMEE* and *JRNN*, respectively. On the *CCKS*2020 dataset, GCN-ED of P is 6.849% and 10.18 higher than BERT-CRF and DMCNN, respectively. EE-GCN integrates syntactic structure and dependent label types and learns and updates graph representations context-dependently, thereby obtaining better extraction capabilities. The GCN-ED method pools the convolution vector of the current word and the entity information in the sentence while integrating syntactic information, thereby improving the extraction performance. In addition, on these datasets, EE-GCN and GCN-ED have higher $F_1$ values than Multi-GAT because the graph attention mechanism focuses on capturing the global details of events in the sentence while ignoring local details.

(3) On the *DuEE* dataset, compared with the Bi-LSTM and *JMEE* methods, the $F_1$ of *PLMEE* increased by 10.082% and 18.641, respectively, and the R-value of BERT-CRF increased by 6.975% and 12.813%, respectively. On the *CCKS*2020 dataset, the *P* and $F_1$ values of *JRNN* increased by 3.155% and 3.694%, respectively, compared with *JMEE*, and by 2.345% and 2.004%, respectively, compared with DMCNN. The possible reason is that the *JRNN* method fully uses the local and global structural information of events in sentences and reduces the error propagation from upstream components to downstream classifiers. At the same time, the interdependence between event triggers and argument roles is strengthened through global features, thereby improving the performance of event extraction.

## 4.4 Ablation experiment

**4.4.1 Effectiveness of different modules.** To verification and evaluation are carried out on the DuEE and CCKS2020 datasets to demonstrate whether the components of the proposed MaskDGNets event extraction framework, such as MaskAM, DGAM, Bi-IndRNN+, and WVEM, play a positive role in the overall framework. The specific experiments are shown in Table 2.

According to Table 2, we draw the following conclusions.

(1) Compared with other modules, the proposed MaskDGNets event extraction method comprises multiple modules and performs best. This shows that the modules work together to enable the network to obtain the optimal feature representation. That is, each module plays a positive role in the model's overall performance. On the *CCKS*2020 dataset, the $F_1$ value of the 'WVEM+MaskAM+Bi-Indrnn+' method is 3.0.43% higher than that of the 'MaskAM + Bi-Indrnn+'. This shows that the word vector embedding module we designed can better map sentences.

**Table 2. Ablation results of different module with our proposed MaskDGNets on the DuEE and CCKS2020 datasets.**

| Models | DuEE | | | CCKS2020 | | |
|---|---|---|---|---|---|---|
| | $P$ | $R$ | $F_1$ | $P$ | $R$ | $F_1$ |
| Bi-Indrnn+ | 62.094 | 63.613 | 62.845 | 70.3 | 70.224 | 70.262 |
| MaskAM | 64.934 | 67.89 | 66.379 | 70.989 | 70.547 | 70.767 |
| DGAM | 74.234 | 70.676 | 72.411 | 72.898 | 75.437 | 74.146 |
| DGAM+MaskAM | 74.76 | 74.53 | 74.645 | 73.325 | 76.239 | 74.754 |
| MaskAM + Bi-Indrnn+ | 75.876 | 75.722 | 75.799 | 74.701 | 76.878 | 75.774 |
| DGAM+Bi-Indrnn+ | 77.413 | 76.076 | 76.739 | 77.834 | 76.968 | 77.399 |
| WVEM+MaskAM+Bi-Indrnn+ | 76.758 | 79.183 | 77.952 | 79.022 | 78.613 | 78.817 |
| MaskAM+DGAM+Bi-Indrnn+ | 81.395 | 80.009 | 80.696 | 79.051 | 80.642 | 79.839 |
| **MaskDGNets** | **84.363** | **78.718** | **81.443** | **86.616** | **88.162** | **87.382** |

Bold font indicates the optimal performance. 'MaskAM' indicates the mask attention module. 'DGAM' indicates the dynamic graph aggregation module. 'WVEM' indicates the word vectors embedding module. 'Bi-Indrnn+' indicates the improved bidirectional independent recurrent neural network module.

(2) On the DuEE baseline dataset, the $F_1$ value of the 'DGAM+Bi-Indrnn+' method is 13.894% and 4.328% higher than that of 'Bi-Indrnn+' and 'DGAM,' respectively. This shows that both modules play a positive role in the model. At the same time, the 'Bi-Indrnn+' module models the overall and contextual semantics of event sentences from both positive and negative directions. It obtains the hidden associations between events through the dynamic graph aggregation module, thereby improving the event extraction performance. The $P$ and $F_1$ of 'DGAM' are 10.14% and 9.566% higher than those of 'Bi-Indrnn+', respectively. This shows that better acquisition of the association between events and related attributes and between events is more beneficial to the overall performance.

(3) On the DuEE and CCKS2020 datasets, the $F_1$ value of the 'MaskAM+DGAM+Bi-Indrnn+' method is 2.744% and 1.022% higher than that of the 'WVEM+MaskAM+Bi-Indrnn+' method, respectively. This shows that DGAM is better at improving model performance. The possible reason is that the dynamic graph nodes transmit and aggregate node information in the neighborhood, enhance the local feature representation, and improve the representation effect of the global and contextual detail semantics. In addition, on the DuEE dataset, the $P$ and $R$ values of the 'DGAM+Bi-Indrnn+' method are improved by 1.537% and 0.354%, respectively, compared with those of 'MaskAM + Bi-Indrnn+'. This shows that 'DGAM' is more beneficial than 'MaskAM' in improving the model's overall performance.

**4.4.2 Effectiveness of reconstruction loss function.** To verify whether the reconstruction loss function plays a positive role in the proposed framework, experimental evaluation is carried out on two baseline datasets, DuEE and CCKS2020. The specific experimental results are shown in Table 3.

From Table 3, we can draw the following conclusions:

(1) The new weighted loss function we designed, through the primary loss function and multiple auxiliary loss functions acting together on the proposed MaskDGNets event extraction framework, can significantly improve the model's overall performance. For example, the $F_1$ of the $L_a + L_b + L_c + L_m$ method on the DuEE dataset is 0.569% and 0.891% higher than that of $L_b + L_c + L_m$ and $L_a + L_b + L_m$, respectively. This shows that the collaborative work of multiple loss functions can enable the network to obtain the optimal feature representation. At the same time, using a separate loss function to supervise each module can effectively suppress the class imbalance problem in the datasets. In particular, on the two baseline datasets, the $F_1$ ratio of $L_b + L_c + L_m$ is 0.322% and 0.434% higher than $L_b + L_c + L_m$, respectively. This also shows

**Table 3. Ablation results of reconstruction loss function with proposed MaskDGNets on the DuEE and CCKS2020 datasets.**

| Models | DuEE | | | CCKS2020 | | |
|---|---|---|---|---|---|---|
| | $P$ | $R$ | $F_1$ | $P$ | $R$ | $F_1$ |
| $L_a$ | 80.127 | 80.019 | 80.073 | 85.08 | 85.018 | 85.049 |
| $L_b$ | 80.207 | 80.419 | 80.313 | 85.207 | 85.373 | 85.29 |
| $L_c$ | 80.274 | 80.571 | 80.422 | 85.673 | 85.51 | 85.591 |
| $L_m$ | 80.283 | 80.632 | 80.457 | 85.931 | 85.519 | 85.725 |
| $L_a + L_b + L_m$ | 80.464 | 80.64 | 80.552 | 86.313 | 85.729 | 86.02 |
| $L_b + L_c + L_m$ | 80.765 | 80.982 | 80.874 | 86.595 | 86.314 | 86.454 |
| $\boldsymbol{L_a + L_b + L_c + L_m}$ | **84.363** | **78.718** | **81.443** | **86.616** | **88.162** | **87.382** |

Bold font indicates the optimal performance.

that using a separate loss function for supervised optimization of the dynamic graph aggregation module (DGAM) is beneficial to the aggregation and representation of graph node features, thereby improving the overall accuracy of the model.

(2) It can be seen from the two sets of datasets that the coordinated work of multiple loss functions is more effective than the supervised learning of the model using a single loss function. For example, on the CCKS2020 dataset, the $F_1$ ratio of $L_m$ and $L_c$ of the $L_a + L_b + L_m$ method is 0.295% and 0.429% higher than $L_m$, respectively. This shows that the coordinated work of multiple loss functions is beneficial to improving the model's overall performance. The $F_1$ of the $L_m$ method is 0.144% and 0.384% higher than that of the $L_b$ and $L_a$ methods, respectively. This shows that the primary loss function we designed plays an absolute leading role in the model's overall performance. It may be that taking the mean of the predicted probability can reduce the differences between the same category and further suppress the category imbalance problem in the datasets.

## 4.5 Discussion

To intuitively demonstrate the effectiveness of the proposed MaskDGNets event extraction framework, we provide the changes in the loss functions of different event extraction methods during the training phase. The specific visualization is shown in Fig 2.

According to Fig 2 clearly shows that the loss function of the MaskDGNets framework we proposed performs better in the training and verification phase. The descent process is relatively smooth, and at 50 iterations, its verification loss value is 0.09463, which is the lowest. This proves that the reconstruction loss function we designed plays a vital role in adjusting and optimizing the network and improving feature representation. The efficiency of different event extraction methods is shown in Fig 3.

According to Fig 3, the FLOPs of the proposed MaskDGNets event extraction framework are within an acceptable range, which is 9.8*G*. Compared with other event extraction methods, Bi-LSTM has the best performance in terms of time efficiency, which is 21.8*G*. Similarly, compared with the event extraction method based on topological graphs, the traditional event extraction method has a better competitive advantage. The possible reason is that the representation and calculation of graph node features are relatively complex, reducing the model's reasoning performance. The FLOPs of the BERT-CRF method are worse than those of Multi-GAT. It may be that the network structure of the BERT-CRF method is more profound, and the number of information transmission layers is greater, resulting in slow reasoning speed. In

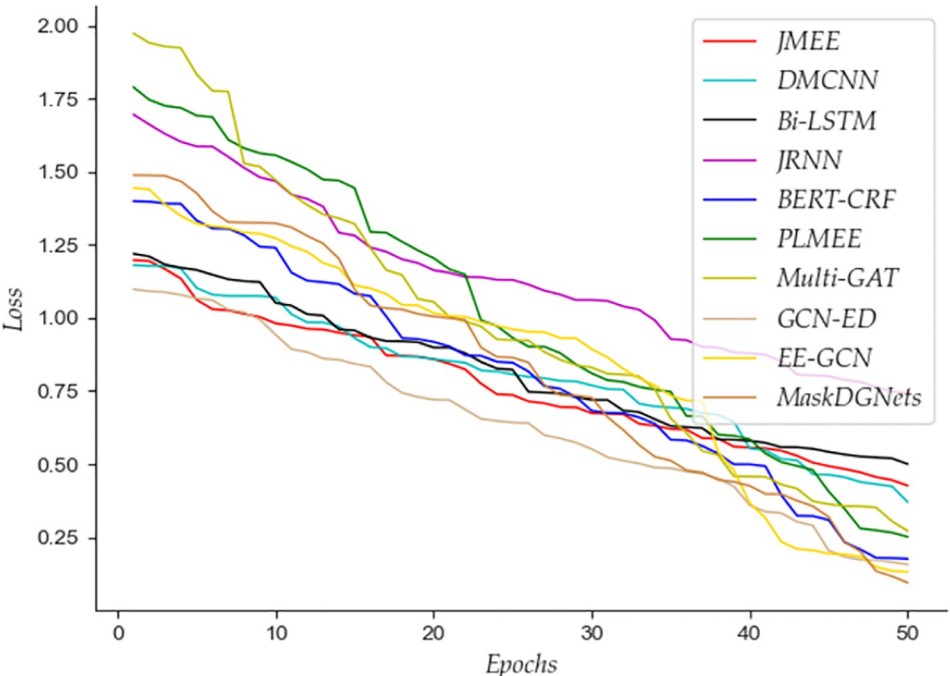

**Fig 2. Validation loss of different event extraction methods.**

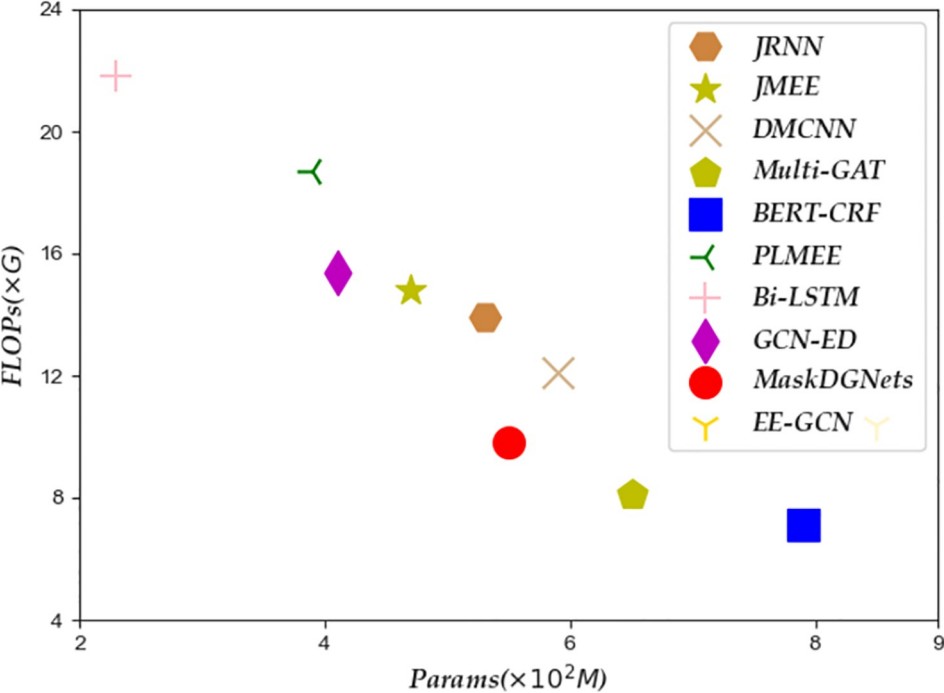

**Fig 3. Efficiency of different event extraction methods.** 'FLOPs' indicates Floating Point Operations Per Second. 'Params' indicates the parameter quantity.

contrast, the Multi-GAT method mainly relies on data-driven to calculate graph node information, reducing the complexity of topological graph construction.

## 5 Conclusions and future research plans

This study considers the shortcomings of traditional deep learning methods in obtaining global, local, and contextual semantics and mining hidden correlations between events and essential attributes, events, and events. It develops a MaskDGNets framework for event extraction. The framework uses the word vector embedding to embed event sentences or texts into low-dimensional space. It uses an improved bidirectional independent recurrent neural network to model global and contextual semantics to improve the performance of semantic representation. Then, masked attention is used to refine the event behavior and related basic attribute detail features, and corresponding weights are assigned to highlight the representation of salient features. In addition, the designed dynamic graph aggregation module is used to establish correlations between events and attributes, events and events, and the interactivity between them is enhanced through node aggregation and transfer functions. Finally, experimental results on two baseline data sets show the framework has good extraction performance and robustness.

Although this framework has achieved superior performance, during the experiment, we found that the reconstruction loss function's learning and balancing factor settings are relatively complex. At the same time, there is still much room for improvement in event extraction efficiency. Therefore, in response to the above limitations, we will develop an efficient and concise semantic guidance network in the future to improve the accuracy of event extraction further while ensuring efficiency.

## Acknowledgments

We are very grateful to the experts for their constructive opinions and to the classmates and teachers who assisted in writing the paper and designing experiments.

## Author Contributions

**Data curation:** Guangwei Zhang.

**Formal analysis:** Guangwei Zhang.

**Investigation:** Fei Xie.

**Methodology:** Lei Yu.

**Supervision:** Fei Xie.

**Validation:** Lei Yu.

**Writing – original draft:** Guangwei Zhang.

**Writing – review & editing:** Guangwei Zhang, Fei Xie.

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
