## [Decision Letter · Decision Letter 0]

8 Oct 2024

PONE-D-24-24636MaskDGNets: Masked-attention guided dynamic graph aggregation network for event extractionPLOS ONE

Dear Dr. Zhang,

Thank you for submitting your manuscript to PLOS ONE. After careful consideration, we feel that it has merit but does not fully meet PLOS ONE’s publication criteria as it currently stands. Therefore, we invite you to submit a revised version of the manuscript that addresses the points raised during the review process.

We look forward to receiving your revised manuscript.

Kind regards,

Ghulam Mustafa, MS Computer Science

Academic Editor

PLOS ONE

3. Thank you for uploading your study's underlying data set. Unfortunately, the repository you have noted in your Data Availability statement does not qualify as an acceptable data repository according to PLOS's standards.

Additional Editor Comments:

Overall study is good please update your manuscript based on reviewer comments. 

Reviewers' comments:

Reviewer's Responses to Questions

**Comments to the Author**

1. Is the manuscript technically sound, and do the data support the conclusions?

Reviewer #1: Yes

Reviewer #2: Yes

2. Has the statistical analysis been performed appropriately and rigorously? 

Reviewer #1: Yes

Reviewer #2: Yes

3. Have the authors made all data underlying the findings in their manuscript fully available?

Reviewer #1: Yes

Reviewer #2: Yes

4. Is the manuscript presented in an intelligible fashion and written in standard English?

Reviewer #1: No

Reviewer #2: Yes

5. Review Comments to the Author

Reviewer #1: 1. Clarify the connection between the MaskDGNets framework and its components, such as MaskAM, DGAM, Bi-IndRNN+, and WVEM. Consider defining these acronyms upon first mention and providing a concise explanation of their individual roles.

2. In the section comparing the MaskDGNets framework to state-of-the-art methods, the analysis could be enriched by discussing more about the underlying reasons why MaskDGNets outperforms the other methods, beyond the broad statement about graph aggregation and bi-directional RNNs. Discuss more on the architectural or design choices that may contribute to its success.

3. The absence of the actual figures mentioned (e.g., Figures 2 and 3) in the provided text makes it harder to follow the conclusions drawn. Ensure that the text clearly references these figures with interpretations that stand alone, even if the figures are not immediately visible.

4. In the discussion of the loss function, provide more context as to why certain loss functions outperform others. For example, if certain combinations of loss functions reduce class imbalance, explain this more fully. This can help reinforce the significance of the loss function choices and justify their inclusion.

5. Proof read the paper there are several gramatical errors.

Reviewer #2: The manuscript was interesting and good written and designed.

Introduction was good written

Design of the study good.

The illustrated figures were good and clear

Discussion was good and including the conclusion of the study.

6. PLOS authors have the option to publish the peer review history of their article (what does this mean?). If published, this will include your full peer review and any attached files.

Reviewer #1: No

Reviewer #2: No

---

## [Author Response · Author response to Decision Letter 0]

24 Oct 2024

Dear Editor：

 Thank you for giving us the opportunity to revise our paper. The following is our response to the specific point that required modification as the editors’ comment.

Response to Editor:

Comment:

We've checked your submission and before we can proceed, we need you to address the following issues:

1. Your Data Availability statement currently reads:

"DuEE: https://www.kaggle.com/datasets/zgw540/maskdgnets-data

CCKS2020:https://www.kaggle.com/datasets/zgw540/maskdgnets-data?select=ccks2020_subtask1

Datasets download account: zgw540@gmail.com

Datasets download password: 15221943277mn"

Our Response to Comment:

 We are very grateful for the expert's constructive comment, and we have made detailed revisions based on it. For instance, We have replaced the data repository and updated the data link, which can be downloaded directly,

DuEE: https://figshare.com/articles/dataset/DuEE1_0_zip/27283248?file=49937835

CCKS2020:https://figshare.com/articles/dataset/ccks2020_zip/27283242?file=49937832

---

## [Editor Report · Decision Letter 1]

29 Oct 2024

MaskDGNets: Masked-attention guided dynamic graph aggregation network for event extraction

PONE-D-24-24636R1

Dear Dr. Zhang,

We’re pleased to inform you that your manuscript has been judged scientifically suitable for publication and will be formally accepted for publication once it meets all outstanding technical requirements.

Kind regards,

Ghulam Mustafa, MS Computer Science

Academic Editor

PLOS ONE
---

## [Editor Report · Acceptance letter]

6 Nov 2024

PONE-D-24-24636R1 

PLOS ONE

Dear Dr. Zhang, 

I'm pleased to inform you that your manuscript has been deemed suitable for publication in PLOS ONE. Congratulations! Your manuscript is now being handed over to our production team.

Kind regards, 

on behalf of

Dr. Ghulam Mustafa 

Academic Editor

PLOS ONE